# Double Burden of Malnutrition: A Population Level Comparative Cross-Sectional Study across Three Sub-Saharan African Countries—Malawi, Namibia and Zimbabwe

**DOI:** 10.3390/ijerph20105860

**Published:** 2023-05-18

**Authors:** Tafadzwa Nyanhanda, Lillian Mwanri, William Mude

**Affiliations:** 1School of Health, Medical and Applied Sciences, Central Queensland University, Melbourne, VIC 4701, Australia; 2College of Sport, Health and Engineering & Institute for Health and Sport, Victoria University, Melbourne, VIC 8001, Australia; 3Research Centre for Public Health, Equity and Human Flourishing, Torrens University Australia, Adelaide Campus, Adelaide, SA 5000, Australia; 4School of Health, Medical and Applied Sciences, Central Queensland University, Cairns City, QLD 4870, Australia

**Keywords:** malnutrition, socio-economic status, obesity, overweight, Africa

## Abstract

Background: The double burden of malnutrition in sub-Saharan African countries at different levels of economic development was not extensively explored. This study investigated prevalence, trends, and correlates of undernutrition and overnutrition among children under 5 years and women aged 15–49 years in Malawi, Namibia, and Zimbabwe with differing socio- economic status. Methods: Prevalence of underweight, overweight, and obesity were determined and compared across the countries using demographic and health surveys data. Multivariable logistic regression was used to ascertain any relationships between selected demographic and socio-economic variables and overnutrition and undernutrition. Results: An increasing trend in overweight/obesity in children and women was observed across all countries. Zimbabwe had the highest prevalence of overweight/obesity among women (35.13%) and children (5.9%). A decreasing trend in undernutrition among children was observed across all countries, but the prevalence of stunting was still very high compared to the worldwide average level (22%). Malawi had the highest stunting rate (37.1%). Urban residence, maternal age, and household wealth status influenced maternal nutritional status. The likelihood of undernutrition in children was significantly higher with low wealth status, being a boy, and low level of maternal education. Conclusions: Economic development and urbanization can result in nutritional status shifts.

## 1. Introduction

One in three persons globally suffer from one or more forms of malnutrition: wasting, stunting, vitamin and mineral deficiency, overweight or obesity, and diet-related non-communicable diseases (NCDs) [1]. The coexistence of undernutrition (wasting and stunting) alongside overnutrition (overweight or obesity) at any population level: country, city, community, household, and individual is defined as the double burden of malnutrition [2]. In 2018, more than 149 million children were stunted, and 40 million were overweight worldwide, with more than a third of the stunted and a quarter of the overweight children living in sub-Saharan Africa [3]. Sub-Saharan Africa is experiencing the double burden of malnutrition with high levels of undernutrition and a growing burden of overweight or obesity and diet-related non-communicable diseases [4]. As a result, the region is not on track to achieve the Sustainable Development Goals of ending hunger and all forms of malnutrition by 2030 [5].

Nutrition transition is a change in dietary activity and body composition patterns that a country goes through on the way to higher levels of economic development, resulting in the coexistence of overnutrition and undernutrition at the population level [6]. Evidence suggests the shifts in dietary and physical patterns in some low- and middle-income countries (LMICs) occur more rapidly than in high-income countries [7]. Studies also showed that urbanization and socio-economic transformation result in increased access to energy-dense and nutrient-poor foods and sedentary jobs, resulting in individuals becoming overweight [8]. In sub-Saharan Africa, the prevalence of obesity is high in urban areas. However, evidence shows that overweight or obesity is increasing faster in rural than in urban areas, even in many LMICs [9]. Ultra-processed foods, which are associated with an increased likelihood of being overweight or obese, are becoming readily available in rural areas forming part of the diets of poor people, including young children [10]. The double burden of malnutrition in sub-Saharan African countries at different levels of economic development was not extensively explored.

Changes in country status from low-income to upper-middle-income led to nutritional shifts resulting in the double burden of malnutrition. The double burden of malnutrition can be observed at the country, household, and even individual level, with the typical pattern being the coexistence of child stunting and wasting and maternal overweight or obesity at the household level [11]. Studies show that in LMICs, the double burden of malnutrition tends to disproportionately affect women than men [12]. Gender inequalities in poverty, high nutritional requirements for pregnancy and lactation make women more vulnerable to the double burden of malnutrition [13]. One study showed a substantial 11.6% increase in the prevalence of overweight and obesity among women of reproductive age in Zimbabwe from 2005 to 2015 [14]. A study analyzing the prevalence and trends of overweight/obesity in women in urbans across 24 countries over the past two-and-a-half decades showed that obesity tripled in Malawi [15]. In a national-level cross-sectional study in Namibia using demographic health survey data, women were preponderant in overweight and obesity [16]. Childhood and adolescence overweight/obesity increased at an alarming rate in sub-Saharan countries. The coexistence of child stunting and wasting and maternal overweight or obesity is not paradoxical, as studies showed that undernutrition in early life predisposes one to obesity and comorbidities later in life if the environment is obesogenic [11]. The burden of malnutrition has serious and lasting developmental, economic, social, and medical impacts for individuals and their families, communities, and countries.

In this study, we compared the trends and prevalence of under- and over nutrition among children aged under five years and women of childbearing age, irrespective of whether the women or children come from the same household, in three selected countries with different levels of gross domestic product (GDP) as a socio-economic development indicator (low income: Malawi; lower-middle-income: Zimbabwe; upper-middle-income: Namibia). We also investigated the relationship between place of residence, wealth quintile, maternal education level, child sex, and age on under- and over nutrition among children and women.

## 2. Materials and Methods

### 2.1. Data Sources

For data on stunting, underweight, and overweight/obesity, we reviewed and analyzed the two latest national demographic and health data from each of the three selected countries covering five years. For information on risk factors for obesity such as food availability and physical activity level, we searched the websites of the Food and Agriculture Organisation of the United Nations, the World Health Organisation, and the World Bank for related data.

### 2.2. Demographic Health Surveys

The data used in this study were derived from the Malawi Demographic and Health Surveys (2010) and (2015–2016) [17,18], the Namibia Demographic and Health Surveys (2005/06) and (2013) [19,20], and the Zimbabwe Demographic and Health Surveys (2010/11) and (2015) [21,22]. The individual women’s data file, which included the women’s BMI measurement, and the household listing data file, which included every child under five years anthropometrics who was in the household at the time of the Demographic and Health Survey interviews, were used for all analyses. Sampling weights were used in calculations as provided by the Demographic and Health Survey.

### 2.3. Outcome Variables

The outcome variables were stunting, wasting, underweight, and overweight/obesity. For children under five years, for all three countries, according to the WHO standard cut-offs [23,24], underweight children were defined as having a weight-for-age, *Z* score *<* −2 standard deviations (SD) below the international reference median, wasting (having a weight-for-height, *Z* score *<* −2 SD), stunting (having a height-for-age, *Z* score *<* −2SD), and overweight (having a weight-for-height *Z*-score *>* +2 SD). Maternal obesity was defined as having BMI ≥30.0 kg/m^2^, overweight (BMI 25.0–29.9 kg/m^2^), and underweight (BMI < 18.5 kg/m^2^). We calculated prevalence by diving the number of people in the sample with the characteristic of interest by the total number of people in the sample.

### 2.4. Independent Variables

The independent variables were grouped into socio-demographic (age, residence, and educational status) and socio-economic (household wealth index).

### 2.5. Food Availability and Physical Activity Levels

Data were derived on country level gross domestic product per capita, demographics, food production, and physical activity levels were collected from multiple sources, including the World Bank, World Development Indicators, the FAO of the UN, Food Balance Sheets, and the WHO. In addition, information on household socio-economic status and individual demographic characteristics was extracted from the survey reports.

### 2.6. Statistical Analysis

SPSS software version 26 (IBM Corp, Armonk, NY, USA) was used to conduct both descriptive and inferential statistics. Multivariable logistic regression models were used to assess the relationship between the three measures of child nutritional status (stunting, underweight, and overweight/obesity), the three measures of women’s BMI (underweight, overweight, and obesity) and maternal education, place of residence, maternal/child age, and household wealth index. Data from each of the three countries were analyzed separately.

## 3. Results

### 3.1. Economic and Demographic Characteristics

Namibia, an upper-middle-income country, had the highest per capita per income, which was 12 times that of Malawi, which had the lowest per capita income (Table 1). Interestingly, Malawi’s food production index was higher than that of Namibia and Zimbabwe. Malawi, which is classified as a low-income country, had the lowest percent (17%) of the urbanized population. Namibia had the highest urban population (51%). Life expectancy was relatively the same across all countries. Malawi had the highest under-5 mortality rate (50 deaths per 1000 live births).

### 3.2. Trends and Prevalence of Undernutrition and Overnutrition in Under-5-Year-Old Children

Significant declining trends of stunting, wasting, and underweight were observed for all countries. Despite this decreasing trend, stunting was highly prevalent in all countries, with Malawi having the highest prevalence (37.1%) and Namibia the lowest (23.73%) in the latest surveys (Table 2). Some rural-urban differences in stunting, wasting, and underweight were observed for all countries (rural to urban ratio ≥ 1.4) (Appendix A).

In general, there was a declining trend in overweight/obesity for Malawi and Namibia. However, for Zimbabwe, there was an increasing trend (Table 2). Zimbabwe had the highest prevalence of overweight/obesity (5.6%), followed by Malawi (4.5%). Comparing the latest survey data, overweight/obesity for children aged under five years was more common in urban areas than rural areas except for Malawi, where the prevalence was the same in both rural and urban areas (rural to urban ratio = 1) (Appendix A). Logistic regression on the most recent survey of the three countries (Table 3) indicates that children living in urban areas were 1.89 (95% CI: 1–3.59, *p* = 0.051) times more likely to experience overweight/obesity than those in rural areas. However, this difference was not statistically significant.

Gender differences in undernutrition and overnutrition were observed. Data from the latest surveys for all countries revealed that male children were more likely to experience undernutrition than female children (Table 3). There was a marked significant gender difference for Namibia in wasting, with boys having a higher odds of wasting than girls (OR = 2.35, 95% CI: 1.51–3.64, *p* < 0.001). A similar result was also observed for Zimbabwe, with male children being 1.33 (95% CI: 1.15–1.54, *p* < 0.001) times more likely of being stunted than female children. Malawian boys were likely to be underweight (OR = 1.29, 95% CI: 1.06–1.57, *p* = 0.013), stunted (OR = 1.19, 95% CI: 1.03–1.38, *p* = 0.016), or overweight/obese (OR = 1.82, 95% CI: 1.29–2.56, *p* = 0.001) compared to female children.

For children under five years old, maternal education influenced the nutritional status of children born in Zimbabwe and Namibia and not in Malawi. For Zimbabwe, the odds ratio of having stunted children were 5.42 (95% CI: 2.54–11.57, *p* < 0.001) times higher for women with no education, 3.21 (95% CI: 1.83–5.65, *p* < 0.001) times higher for women with primary education, and 2.59 (95% CI: 1.5–4.46, *p* = 0.001) times higher for women with secondary education than women who had higher education.

Household wealth status had a significant influence on stunting in children under five years. For all countries, children from the poorest, poorer, and middle quintiles were more likely to be stunted compared to the richest quintile; for Zimbabwe, even those in the rich quintile were also likely to be stunted (Table 3). Malawian children in the poorest quintile were also more likely to be underweight than those in the richest quintile, whereas Namibian children from poor and poorest quintiles were also likely to be underweight (Table 3).

### 3.3. Trends and Prevalence of Undernutrition and Overnutrition in Women of Childbearing Age

Table 3 shows increasing trends of overweight and obesity among women of child bearing age in all the three countries regardless of residency, with the highest prevalence of overweight/obesity observed in Zimbabwe (34.9%), while Malawi had the lowest prevalence (20.7%) in the latest surveys. Declining trends in the prevalence of underweight was observed for all countries, regardless of residency (Table 4). Namibia had the highest prevalence of underweight women (15.25%). A comparison of the overweight-to-underweight ratio for the earliest and latest survey years of all three countries revealed a higher prevalence of overweight/obesity in women of childbearing age was higher than that of underweight, Zimbabwe had a larger difference than other countries, having over five times more women with overweight/obesity than underweight (Table 4).

Rural/urban differences in overweight/obesity prevalence were observed for all countries with higher levels for women in urban areas compared to women in rural areas. Place of residence also influenced the nutritional status of women. Malawian women living in urban areas were 1.99 (95% CI: 1.41–2.80, *p* < 0.001) times likely to be obese and 1.45 (95% CI: 1.12–1.88, *p* = 0.005) times more likely to be overweight than those living in the rural areas (Table 5). No rural/urban difference was observed for Namibia.

For Malawi, maternal education did not influence the nutritional status of the women. In contrast, for Zimbabwe, women whose highest educational attainment was primary or secondary education were less likely to experience obesity than the reference group of higher education attainment (Table 5). Namibian women with no education, primary education, or secondary education were more likely to have wasted than those with higher education.

For all countries, household wealth status had an influence on overnutrition and undernutrition in women. For both Malawi and Zimbabwe women, experiencing overweight or obesity was less likely among the poorest, poorer, middle, and richer quintiles than the reference group of richest quintiles (Table 5). In Namibia, being overweight or obese was less likely among women from the poorest, poorer, and middle quintiles compared to the richest quintile. The wealth status did not influence the likelihood of being underweight among Malawian women of childbearing age. However, for Zimbabwean women, the odds ratio of being underweight was 3.5 (95% CI: 1.99–6.15, *p* < 0.001) times higher among those who are in the poorest quintile compared to those who are in the richest quintile. Namibian women of childbearing age in the poorest quintile were 1.81 (95% CI: 1.14–2.88, *p* = 0.013) times more likely to be underweight than those in the richest quintile.

The women’s age also determined nutritional status. In general, younger women were less likely to be obese or overweight than older women but more likely to be underweight (Table 5). For Malawi, younger women aged between 15–19 years (OR = 0.04, 95% CI:0.02–0.08, *p* < 0.001), 20–24 years (OR = 0.16, 95% CI:0.08–0.29, *p* < 0.0010), and 25–29 years (OR = 0.32, 95% CI:0.18–0.55, *p* < 0.001) were less likely to be obese compared to women aged between 45 and 49 years. Zimbabwean women aged 15–19 years had a higher likelihood of being underweight than women aged 45–49 years (OR = 1.62, 95% CI:1–2.61, *p* = 0.049). For Namibia, younger women aged between 15–19 years (OR = 0.04, 95% CI:0.02–0.08, *p* < 0.001), 20–24 years (OR = 0.08, 95% CI:0.05–0.07, *p* < 0.001), and 25–29 years (OR = 0.22, 95% CI:0.14–0.34, *p* < 0.001) were less likely to be overweight or obese compared to women aged between 45 and 49 years.

### 3.4. Food Availability

Per capita energy supply remained stable, with an increase from 2000 to 2017 for all countries (Table 6). The availability of cereals in Malawi increased by 50%, while the supply decreased by 50% in Zimbabwe. There was a significant increase in meat availability in Malawi. Fruit and vegetable availability in Malawi more than quadrupled, followed by Namibia, which had a 50% increase.

### 3.5. Physical Activity Levels

Countries with higher economic status had an increased prevalence of inadequate physical activity. Namibia had the highest prevalence of inadequate physical activity (33.4%), followed by Zimbabwe (26.8%). Malawi had the lowest prevalence of physical inactivity (15.6%) (Table 6).

## 4. Discussion

We compared the prevalence of undernutrition and overnutrition among children under five years and women of childbearing age in three selected sub-Saharan African countries with varying levels of socio-economic development. Findings were based on data from national surveys, World Bank development indicators, food balance sheets (FAO), and WHO physical activity levels data. Large disparities in nutritional status among young children and women were observed across the three countries. Rural–urban and gender differences were also observed in the prevalence of overnutrition and undernutrition. Overweight/obesity was more common in urban areas than in rural areas.

Many studies showed that socio-economic development and increased urbanization resulted in diets shifts, leading to the increasing burdens of overweight or obesity and diet-related noncommunicable diseases, such as diabetes and heart disease [27]. With increasing socio-economic development, there is also the growth of the urban population. Urban populations tend to consume more calories due to the availability of foods high in saturated and trans fats, refined carbohydrates, simple sugars, salt, animal source food, and processed foods and reduced consumption of traditional starchy carbohydrates as dietary staples [27]. This study found that countries with higher economic status had a higher prevalence of overweight/obesity in childbearing women. Neuman et al., 2014, found a positive association between GDP and mean BMI, though non-significant [28]. However, Namibia, an upper-middle-income country, had the second-lowest prevalence of overweight/obesity in children and the highest prevalence of underweight women. This highlights that besides socio-economic transformation and urbanization, other factors contribute to overweight/obesity in these countries [27]. The perception of a larger body size as a sign of affluence and desiring women with larger body sizes in some African cultures might explain the observed differences in the prevalence of overweight/obesity in these countries [29]. Physical inactivity is linked to an increased prevalence of overweight or obesity and NCDs. This study found that countries with higher economic status had an increased prevalence of inadequate physical activity. This finding is in line with previous studies. For example, one study in Cameroon observed that urban residence compared with rural residence was associated with lower physical activity energy expenditure and higher prevalence of metabolic syndrome [30].

We found rural–urban differences in the prevalence of overweight/obesity were observed with higher levels in urban areas compared to rural areas. In urban areas, markets are increasingly replacing fresh produce and selling commercially prepared and processed foods from transnational and local industries and street vendors [31]. However, it should be noted that increasing trends of overnutrition in both rural and urban areas were observed in all countries. Bixby et al., 2019, found that particularly in LMICs, BMI is rising at the same rate or faster in rural areas compared to urban areas, except among women in sub-Saharan Africa, highlighting that urban living and urbanization may not be the only key driver of the global epidemic of obesity.

Undernutrition in children aged under five years is highly prevalent in the three selected sub-Saharan countries, with stunting being the most prevalent. Rural–urban differences in stunting, wasting, and underweight were observed in most of the countries. Malawi, the country with the lowest gross domestic product per capita, had the highest prevalence of stunting (37.1%) and infant mortality rate. In sub-Saharan Africa, nutritional status was found to be a central determinant of child mortality [32].

In this study, maternal education had a limited influence on women’s nutritional status. However, a significant influence was observed for children under five years old. Maternal education on child health and nutrition was well demonstrated to play a significant role in many studies. Higher maternal educational attainment was shown to improve the socioeconomic status of mothers leading to better children’s health and nutritional outcomes [33]. Higher socioeconomic status can result in better feeding practices and the utilization of health services, and maternal education improves the mother’s knowledge about child health, including causes, prevention, and treatment of diseases [34]. Our findings show the influence of wealth status on overnutrition and undernutrition, with those in the poorest quintile more likely to be undernourished or have stunted children than those in the richest quintile, and those in the richest quintile were more likely to experience overweight/obese than those in the poorest quintile.

The production of food supply increased in most countries, with a significant increase in meat production in Malawi. Meat availability was found to correlate positively with obesity prevalence [35]. However, general market availability or supply of food does not necessarily translate to consumption.

The study has some important limitations. First, the finding cannot be entirely generalizable to the subcontinent, as the three countries included in the study might not be necessarily like the remaining sub-Saharan African countries. Second, we used data from cross-sectional surveys; therefore, causal inference is not viable, and the available data were from different years, which limited our comparisons. However, this study used the most recent national data and compared selected countries based on their economic status, representing Southern Africa, and the findings may only reflect this region.

## 5. Conclusions

The double burden of malnutrition in sub-Saharan Africa is of critical concern. Current data indicate that the world is not on track to achieve the United Nations’ Sustainable Development Goal 2: Zero Hunger by 2030, which is concerning. In recent years, many health-related policies and interventions in Africa focused on addressing undernutrition and infectious disease; however, the current nutrition status trends highlight the need to address all forms of malnutrition. Policies and interventions to address malnutrition in sub-Saharan Africa and other transitional societies need to be double-pronged and gender-sensitive. There is no one-size-fits-all solution for countries, and policymakers will need to assess the context-specific barriers. When formulating and implementing national policies or interventions, factors such as national economic development, urbanization, food availability, diet quality, and physical activity levels need to be considered.

## Figures and Tables

**Table 1 ijerph-20-05860-t001:** Economic and Demographic Characteristics.

Country	Income Level	GDP per Capita (US $) *	Life Expectancy (y) *	Under 5 Mortality *	Population (Thousands) †	Urban Population (% of Total Population) *	Food Production Index †
Malawi	Low	411.6	64	50	18,628.75	17	147.2
Zimbabwe	Lower-middle	1464.00	61	46	14,645.47	32	88.8
Namibia	Upper-middle	4957.50	63	40	2494.53	51	92.6

* World Bank development indicators: GDP (2019), Life expectancy (2018), Under-5 mortality (2018), Population (2019), Urban Population (2019). † The World Bank Food production index covers food crops that are considered edible and that contain nutrients. Coffee and tea are excluded because, although edible, they have no nutritive value.

**Table 2 ijerph-20-05860-t002:** Prevalence and trends of overnutrition and undernutrition among children under 5 years.

Country		2010	2016	Nature of Trend
	Stunting (%, (95%CI))			
Malawi	All	47.1 (45.23, 48.98)	37.11 (35.5, 38.76)	
	Urban	40.69 (35.01, 46.62)	25.01 (20.61, 30)	
	Rural	48.22 (46.28, 50.17)	38.86 (37.16, 40.6)	
	Boys	51.14 (48.52, 53.75)	38.97 (36.68, 41.31)	
	Girls	43.26 (40.73, 45.83)	35.35 (33.17, 37.6)	
	Wasting (%, (95%CI))			
	All	3.98 (3.34, 4.74)	2.76 (2.3, 3.31)	
	Urban	2.36 (1.25, 4.43)	3.33 (1.91, 5.75)	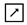
	Rural	4.27 (3.56, 5.1)	2.68 (2.21, 3.23)	
	Boys	4.22 (3.35, 5.29)	3.32 (2.61, 4.21)	
	Girls	3.76 (2.9, 4.86)	2.22 (1.68, 2.93)	
	Overweight/Obesity (%, (95%CI))		
	All	8.26 (7.33, 9.3)	4.55 (3.91, 5.29)	
	Urban	8.9 (6.44, 12.17)	4.66 (2.9, 7.4)	
	Rural	8.15 (7.16, 9.27)	4.54 (3.87, 5.32)	
	Boys	9.1 (7.72, 10.69)	5.84 (4.8, 7.09)	
	Girls	7.47 (6.27, 8.89)	3.33 (2.61, 4.24)	
		2006/07	2013	
	Stunting (%, (95%CI))			
Namibia	All	28.96 (27.46, 30.51)	23.73 (21.79, 25.78)	
	Urban	23.78 (21.02, 26.78)	16.7 (13.95, 19.87)	
	Rural	31.44 (29.69, 33.24)	27.81 (25.3, 30.48)	
	Boys	31.48 (29.39, 33.66)	26.53 (23.67, 29.59)	
	Girls	26.45 (24.41, 28.59)	20.94 (18.45, 23.67)	
	Wasting (%, (95%CI))			
	All	7.46 (6.68, 8.32)	6.22 (5.22, 7.41)	
	Urban	5.62 (4.4, 7.15)	4.98 (3.56, 6.92)	
	Rural	8.34 (7.38, 9.41)	6.95 (5.66, 8.51)	
	Boys	7.28 (6.23, 8.49)	8.58 (6.97, 10.51)	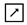
	Girls	7.64 (6.56, 8.87)	3.88 (2.84, 5.28)	
	Overweight/Obesity (%, (95%CI))		
	All	4.32 (3.68, 5.05)	3.41 (2.68, 4.32)	
	Urban	7.43 (5.91, 9.3)	4.15 (2.83, 6.04)	
	Rural	2.82 (2.29, 3.48)	2.98 (2.2, 4.02)	-
	Boys	4.47 (3.56, 5.61)	3.12 (2.22, 4.35)	
	Girls	4.16 (3.34, 5.16)	3.7 (2.64, 5.15)	
		2010/11	2015	
	Stunting (%, (95%CI))			
Zimbabwe	All	31.96 (30.56, 33.39)	26.76 (25.43, 28.12)	
	Urban	27.52 (24.82, 30.39)	22.11 (19.75, 24.67)	
	Rural	33.42 (31.8, 35.08)	28.49 (26.92, 30.12)	
	Boys	35.72 (33.7, 37.79)	29.51 (27.65, 31.43)	
	Girls	28.27 (26.4, 30.21)	24.05 (22.27, 25.92)	
	Wasting (%, (95%CI))			
	All	2.97 (2.52, 3.51)	3.17 (2.7, 3.73)	-
	Urban	2.14 (1.35, 3.36)	2.39 (1.64, 3.48)	-
	Rural	3.25 (2.72, 3.87)	3.47 (2.89, 4.15)	-
	Boys	3.59 (2.88, 4.46)	3.19 (2.53, 4.01)	-
	Girls	2.37 (1.83, 3.05)	3.16 (2.53, 3.95)	-
	Overweight/Obesity (%, (95%CI))			
	All	5.49 (4.84, 6.22)	5.59 (4.96, 6.3)	-
	Urban	5.42 (4.25, 6.89)	7.55 (6.2, 9.17)	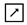
	Rural	5.51 (4.76, 6.37)	4.86 (4.17, 5.65)	
	Boys	6.29 (5.34, 7.4)	6.12 (5.2, 7.21)	-
	Girls	4.7 (3.88, 5.69)	5.06 (4.25, 6.02)	-

- No change (less than a 0.5% change), 
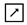
 Increase, 

 Decrease.

**Table 3 ijerph-20-05860-t003:** Multivariable Logistic Regression analysis of under- and overnutrition among children under 5 years.

Year	Variable Categories	Stunting OR (95% CI)	Underweight OR (95% CI)	Wasting OR (95% CI)	Overweight/Obesity OR (95% CI)
Malawi 2016	Urban	0.85 (0.61, 1.18)	1.06 (0.63, 1.77)	1.06 (0.59, 1.92)	0.77 (0.45, 1.33)
	Rural (Ref)				
	Poorest	2.21 (1.62, 3.01) ‡	2.08 (1.28, 3.37) †	1.01 (0.52, 1.96)	0.63 (0.33, 1.22)
	Poorer	1.8 (1.32, 2.44) ‡	1.72 (1.06, 2.8) *	0.88 (0.47, 1.67)	0.9 (0.47, 1.71)
	Middle	1.57 (1.14, 2.15) †	1.67 (0.99, 2.8)	1.1 (0.57, 2.09)	0.86 (0.46, 1.6)
	Richer	1.25 (0.92, 1.71)	1.32 (0.81, 2.16)	0.8 (0.42, 1.56)	0.86 (0.47, 1.57)
	Richest (Ref)				
	Male	1.19 (1.03, 1.38) *	1.29 (1.06, 1.57) †	1.44 (0.98, 2.12)	1.82 (1.29, 2.56) ‡
	Female (Ref)				
	No Education	2.79 (1.06, 7.34) *	3.75 (0.95, 14.81)	0.45 (0.08, 2.65)	0.41 (0.11, 1.52)
	Primary	2.54 (0.98, 6.59)	3.21 (0.83, 12.37)	0.53 (0.1, 2.85)	0.42 (0.14, 1.28)
	Secondary	2.27 (0.87, 5.91)	2.9 (0.76, 11.06)	0.56 (0.1, 2.96)	0.35 (0.12, 1.05)
	Higher (Ref)				
	Child Age	1.01 (1.01, 1.02) ‡	1.01 (1, 1.02) †	0.98 (0.97, 1)	0.96 (0.94, 0.97) ‡
Namibia 2013	Urban	0.98 (0.7, 1.38)	0.82 (0.55, 1.24)	0.92 (0.54, 1.56)	0.91 (0.49, 1.68)
	Rural (Ref)				
	Poorest	3.28 (1.64, 6.55) ‡	3.29 (1.25, 8.71) *	1.56 (0.56, 4.38)	0.55 (0.12, 2.41)
	Poorer	3.03 (1.56, 5.89) ‡	2.91 (1.14, 7.44) *	1.32 (0.5, 3.53)	0.55 (0.14, 2.16)
	Middle	2.5 (1.31, 4.75) †	2.48 (0.95, 6.45)	1.27 (0.47, 3.41)	1.04 (0.3, 3.62)
	Richer	1.89 (0.99, 3.59)	2.02 (0.77, 5.31)	0.72 (0.26, 1.98)	1.44 (0.41, 4.99)
	Richest (Ref)				
	Male	1.2 (0.92, 1.57)	1.33 (0.95, 1.84)	2.35 (1.51, 3.64) ‡	0.65 (0.35, 1.18)
	Female (Ref)				
	No Education	2.47 (0.8, 7.67)	2.16 (0.51, 9.03)	30.56 (3.71, 251.77) †	0.39 (0.05, 3.12)
	Primary	2.14 (0.72, 6.39)	1.76 (0.44, 7.02)	11.36 (1.43, 90.19) *	0.68 (0.15, 3.1)
	Secondary	1.54 (0.53, 4.5)	1.22 (0.31, 4.75)	11.13 (1.46, 85.08) *	0.63 (0.15, 2.69)
	Higher (Ref)				
	Child Age	1.01 (1.01, 1.02) ‡	1.01 (1, 1.02) *	0.96 (0.94, 0.97) ‡	1.01 (0.99, 1.03)
Zimbabwe 2015	Urban	1.26 (0.87, 1.82)	0.81 (0.44, 1.49)	1.3 (0.58, 2.9)	1.89 (1, 3.59) *
	Rural				
	Poorest	2.34 (1.47, 3.74) ‡	1.47 (0.65, 3.31)	2.8 (0.99, 7.91)	1.35 (0.59, 3.07)
	Poorer	2.01 (1.26, 3.2) †	1.43 (0.63, 3.24)	2.97 (1.04, 8.43) *	1.19 (0.52, 2.72)
	Middle	1.82 (1.14, 2.91) †	1.18 (0.51, 2.7)	2.06 (0.71, 6.01)	1.3 (0.56, 3.01)
	Richer	1.72 (1.27, 2.32) ‡	1.36 (0.76, 2.41)	1.93 (0.86, 4.33)	1.09 (0.68, 1.76)
	Richest				
	Male	1.33 (1.15, 1.54) ‡	1.18 (0.94, 1.49)	1.04 (0.74, 1.47)	1.22 (0.93, 1.61)
	Female				
	No Education	5.42 (2.54, 11.57) ‡	3.72 (0.91, 15.26)	0.4 (0.07, 2.32)	0.34 (0.07, 1.55)
	Primary	3.21 (1.83, 5.65) ‡	2.94 (1.06, 8.18) *	0.76 (0.27, 2.18)	0.59 (0.31, 1.13)
	Secondary	2.59 (1.5, 4.46) ‡	2.43 (0.89, 6.64)	0.56 (0.2, 1.55)	0.65 (0.36, 1.17)
	Higher				
	Child Age	1 (1, 1)	0.99 (0.99, 1) †	0.97 (0.96, 0.98) ‡	0.96 (0.95, 0.97) ‡

Note: Dependent Variable: Stunting, Underweight, Wasting, Overweight/Obesity [reference category = Not stunted (z-score is ≥−2 standard deviations), Not underweight (z-score is ≥−2 standard deviations), Not wasted (z-score is ≥−2 standard deviations), Not overweight/obese (z-score is −2 to 2 standard deviations)], ‡ *p* ≤ 0.001 † *p* ≤ 0.01 * *p* ≤ 0.05.

**Table 4 ijerph-20-05860-t004:** Prevalence of under- and overnutrition among women of childbearing age.

Country		2010	2015/16	Nature of Trend
	Underweight BMI < 18.5 kg/m^2^ (95% CI)
Malawi	All	8.63 (7.83, 9.51)	7.15 (6.44, 7.94)	
	Urban	7.28 (5.47, 9.64)	6.21 (4.49, 8.54)	
	Rural	8.98 (8.11, 9.93)	7.36 (6.6, 8.21)	
	Rural/Urban Ratio	1.23 (1.48, 1.03)	1.19 (1.47, 0.96)	
	Overweight/Obesity BMI ≥ 25 Kg/m^2^ (95% CI)
	All	17.09 (15.49, 18.86)	20.9 (19.19, 22.74)	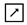
	Urban	28.08 (23, 34.11)	36.63 (31.01, 43.02)	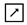
	Rural	14.28 (12.85, 15.88)	17.4 (15.81, 19.14)	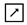
	Rural/Urban Ratio	0.51 (0.56, 0.47)	0.48 (0.51, 0.44)	
		2006/07	2013	
	Underweight BMI < 18.5 kg/m^2^ (95% CI)
Namibia	All	15.25 (14.41, 16.13)	13.88 (12.68, 15.18)	
	Urban	11.38 (10.2, 12.69)	10.63 (9.11, 12.36)	
	Rural	18.77 (17.62, 19.97)	17.71 (15.91, 19.67)	
	Rural/Urban Ratio	1.65 (1.73, 1.57)	1.67 (1.75, 1.59)	
	Overweight/Obesity BMI ≥ 25 Kg/m^2^ (95% CI)
	All	28.43 (26.7, 30.26)	31.51 (28.94, 34.25)	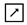
	Urban	37.9 (34.86, 41.14)	39.73 (35.7, 44.09)	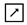
	Rural	19.82 (18.17, 21.61)	21.82 (19.16, 24.81)	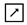
	Rural/Urban Ratio	0.52 (0.52, 0.53)	0.55 (0.54, 0.56)	
		2010/11	2015	
	Underweight BMI < 18.5 kg/m^2^ (95% CI)
Zimbabwe	All	7.07 (6.48, 7.7)	6.08 (5.54, 6.65)	
	Urban	5.22 (4.38, 6.21)	4.06 (3.42, 4.83)	
	Rural	8.2 (7.43, 9.04)	7.31 (6.57, 8.13)	
	Rural/Urban Ratio	1.57 (1.7, 1.46)	1.8 (1.92, 1.68)	
	Overweight/Obesity BMI ≥ 25 Kg/m^2^ (95% CI)
	All	31.34 (29.64, 33.13)	35.13 (33.36, 36.97)	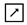
	Urban	40.95 (37.84, 44.26)	46.59 (43.36, 49.97)	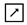
	Rural	25.44 (23.52, 27.5)	28.09 (26.12, 30.2)	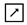
	Rural/Urban Ratio	0.62 (0.62, 0.62)	0.6 (0.6, 0.6)	

(less than a 0.5% change) 
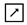
 Increase 

 Decrease.

**Table 5 ijerph-20-05860-t005:** Multivariable Logistic Regression analysis of under- and overnutrition among women of childbearing age.

		Underweight BMI < 18.5 Kg/m^2^	Overweight BMI 25–29.9 Kg/m^2^	Obese BMI ≥ 30 Kg/m^2^
**Year**	**Variable categories**	**OR (95% CI)**	**OR (95% CI)**	**OR (95% CI)**
Malawi 2015/16	15–19	1.86 (1.1, 3.13)	0.23 (0.16, 0.34) ‡	0.04 (0.02, 0.08) ‡
	20–24	0.74 (0.42, 1.29)	0.43 (0.3, 0.63) ‡	0.16 (0.08, 0.29) ‡
	25–29	0.84 (0.48, 1.49)	0.75 (0.52, 1.07)	0.32 (0.18, 0.55) ‡
	30–34	1.19 (0.68, 2.1)	0.95 (0.67, 1.36)	0.85 (0.51, 1.41)
	35–39	1.03 (0.58, 1.83)	1.05 (0.74, 1.5)	0.91 (0.54, 1.51)
	40–44	1.1 (0.61, 1.98)	0.85 (0.58, 1.26)	0.82 (0.47, 1.43)
	45–49 (Ref)			
	Urban	1.21 (0.77, 1.91)	1.45 (1.12, 1.88) †	1.99 (1.41, 2.8) ‡
	Rural (Ref)			
	No Education	3.3 (0.6, 18.25)	1.15 (0.66, 1.99)	0.81 (0.37, 1.76)
	Primary	3.09 (0.57, 16.73)	0.9 (0.55, 1.49)	0.76 (0.38, 1.5)
	Secondary	2.58 (0.47, 14.22)	1.02 (0.64, 1.64)	1 (0.51, 1.95)
	Higher (Ref)			
	Poorest	1.08 (0.69, 1.69)	0.36 (0.26, 0.5) ‡	0.14 (0.07, 0.28) ‡
	Poorer	0.84 (0.52, 1.36)	0.36 (0.27, 0.49) ‡	0.25 (0.15, 0.42) ‡
	Middle	0.92 (0.58, 1.46)	0.54 (0.4, 0.72) ‡	0.25 (0.15, 0.42) ‡
	Richer	1.05 (0.67, 1.65)	0.68 (0.52, 0.89) †	0.47 (0.33, 0.68) ‡
	Richest (Ref)			
Namibia 2013	15–19	1.97 (1.27, 3.06) *	0.12 (0.07, 0.19) ‡	0.04 (0.02, 0.07) ‡
	20–24	1.17 (0.73, 1.86)	0.28 (0.19, 0.42) ‡	0.08 (0.05, 0.14) ‡
	25–29	0.8 (0.48, 1.34)	0.52 (0.35, 0.76) ‡	0.22 (0.14, 0.34) ‡
	30–34	0.64 (0.37, 1.11)	0.79 (0.53, 1.17)	0.54 (0.35, 0.83) †
	35–39	1.06 (0.62, 1.81)	0.84 (0.57, 1.25)	0.65 (0.42, 1.02)
	40–44	1.25 (0.72, 2.15)	1.18 (0.77, 1.79)	1.29 (0.83, 1.99)
	45–49 (Ref)			
	Urban	1 (0.75, 1.34)	1.19 (0.93, 1.53)	1.29 (0.97, 1.72)
	Rural (Ref)			
	No Education	1.34 (0.68, 2.66)	1 (0.56, 1.8)	1.24 (0.64, 2.41)
	Primary	1.71 (0.98, 3)	1.02 (0.65, 1.61)	0.82 (0.49, 1.38)
	Secondary	1.18 (0.7, 1.98)	1.02 (0.7, 1.49)	1.08 (0.71, 1.66)
	Higher (Ref)			
	Poorest	1.81 (1.14, 2.88) †	0.32 (0.2, 0.49) ‡	0.06 (0.03, 0.13) ‡
	Poorer	1.43 (0.92, 2.21)	0.42 (0.29, 0.62) ‡	0.17 (0.11, 0.28) ‡
	Middle	1.46 (0.94, 2.27)	0.66 (0.47, 0.94) *	0.38 (0.26, 0.56) ‡
	Richer	1.39 (0.94, 2.06)	1.1 (0.81, 1.49)	0.84 (0.6, 1.19)
	Richest (Ref)			
Zimbabwe 2015	15–19	1.62 (1, 2.61) *	0.22 (0.16, 0.29) ‡	0.03 (0.02, 0.05) ‡
	20–24	0.87 (0.52, 1.45)	0.36 (0.27, 0.48) ‡	0.1 (0.07, 0.15) ‡
	25–29	0.46 (0.26, 0.8) †	0.58 (0.44, 0.77) ‡	0.24 (0.17, 0.33) ‡
	30–34	0.74 (0.43, 1.27)	0.9 (0.68, 1.2)	0.43 (0.32, 0.59) ‡
	35–39	0.84 (0.48, 1.48)	0.85 (0.64, 1.14)	0.67 (0.49, 0.93) *
	40–44	0.65 (0.36, 1.2)	1.12 (0.83, 1.51)	0.81 (0.59, 1.12)
	45–49 (Ref)			
	Urban	1.73 (1.08, 2.76) *	1 (0.79, 1.26)	0.97 (0.73, 1.29)
	Rural (Ref)			
	No Education	0.77 (0.25, 2.36)	0.9 (0.49, 1.64)	0.56 (0.22, 1.4)
	Primary	0.73 (0.39, 1.38)	0.88 (0.66, 1.17)	0.63 (0.46, 0.87) †
	Secondary	0.79 (0.43, 1.45)	0.9 7(0.76, 1.25)	0.77 (0.6, 1) *
	Higher (Ref)			
	Poorest	3.5 (1.99, 6.15) ‡	0.36 (0.26, 0.49) ‡	0.09 (0.05, 0.13) ‡
	Poorer	3.42 (1.95, 6) ‡	0.46 (0.34, 0.63) ‡	0.17 (0.12, 0.26) ‡
	Middle	3.3 (1.89, 5.76) ‡	0.6 (0.45, 0.81) ‡	0.25 (0.17, 0.37) ‡
	Richer	1.7 (1.19, 2.43) †	0.82 (0.67, 1) *	0.54 (0.43, 0.69) ‡
	Richest (Ref)			

Note: Dependent Variable: BMI categories (reference category = Normal BMI 18.5–24.9 kg/m^2^), ‡ *p* ≤ 0.001 † *p* ≤ 0.01 * *p* ≤ 0.05.

**Table 6 ijerph-20-05860-t006:** Time trends of the food supply and physical activity levels of the three selected countries.

Country	Income Level	Per Capita Total Energy Supply per Day ^1^		Cereals (1000 Tonnes) ^1^		Meat (1000 Tonnes) ^1^		Fruits and Vegetables (1000 Tonnes) ^1^		Pulses (1000 Tonnes) ^1^		Prevalence of Inadequate Physical Activity (%) ^2^
		2000	2017	2000	2017	2000	2017	2000	2017	2000	2017	2016
Malawi	Low	2204	2647	2607	3711	63	412	1030	4818	250	748	15.6 [13.3–17.9]
Namibia	Upper-middle	2246	2431	121	127	97	77	39	134	13	23	33.4 [26.4–41.1]
Zimbabwe	Lower-middle	1979	2173	2519	1121	189	257	380	526	52	27	26.8 [20.5–34.3]

^1^ Data source: FAO of the UN: Food Balance Sheets—figures are amounts produced domestically [25]. ^2^ Prevalence of insufficient physical activity among adults aged 18+ years (age-standardized estimate); inadequate physical activity according to the WHO recommendation is defined as a population attaining <150 min of moderate-intensity physical activity per week, or <75 min of vigorous-intensity physical activity per week, or equivalent [26].

## Data Availability

DHS data for the three countries are available Measure DHS.

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
