# Peer review of "Double Burden of Malnutrition: A Population Level Comparative Cross-Sectional Study across Three Sub-Saharan African Countries—Malawi, Namibia and Zimbabwe"

_ijerph, 2023, doi:10.3390/ijerph20105860_

Round 1

Reviewer 1 Report

I suggest adding Malawi, Namibia and Zimbabwe to the title.

In the conclusion, it is reasonable to discuss the prospects and what needs to be done to achieve the Sustainable Development Goals in this area.

It is interesting to link eating habits at the community level to urban practices.

Author Response

We thank reviewer 1 for their compliments. We have attached our responses for their consideration.

Reviewer 2 Report

Dear Authors, all comments and suggestions have been made on the main manuscript. It will be attached here.

However, many clarifications are needed, and I will include some recommended articles to assist you in your corrections. Perhaps, citations of them are highly recommended.

Author Response

We thank reviewer 2 for their feedback. We have attached our responses for their consideration and directed them to appropriate information in the article for information they might have missed. 

Reviewer 3 Report

Major issues

Please indicate funding sources and conflicts of interest if any.

Indicate if approval from Institutional Review Board(s) was obtained.

 Minor issues

1)      Table 2: Nature of trend. If the authors are looking to report linear trends, at least three observations would be required to make such as inference. Unfortunately, they report two observations in the table and in table 4 and display arrows to indicate increasing or decreasing trends. This depiction has the potential to mislead the reader. I suggest the caption be changed to read “Change from baseline” and the arrows defined to mean “increased” or “decreased” change as applicable.

2)      Table 2: please define the prevalence calculation. The same applies to Table 4.

3)      Table 3: I suspect the authors are reporting adjusted odds ratios. Please state this in the table title.

4)      Whereas the authors used country-level data collected and properly weighted by the various government health departments, they elected to use data available from the website of international organizations, which may lack the level of granularity needed for proper analysis. The authors need to acknowledge the limitations of BMI to assess weight-related risk. These limitations need to be discussed.

Author Response

We thank reviewer 3 for their reviews. We have attached our responses for the reviewer to consider.
